# Phat Queens Emerge Fashionably Late: Body Size and Condition Predict Timing of Spring Emergence for Queen Bumble Bees

**DOI:** 10.3390/insects13100870

**Published:** 2022-09-24

**Authors:** Ellen C. Keaveny, Michael E. Dillon

**Affiliations:** 1Department of Zoology and Physiology, University of Wyoming, Laramie, WY 82071, USA; 2Program in Ecology, University of Wyoming, Laramie, WY 82071, USA

**Keywords:** body mass, lipids, body shape, *Bombus*

## Abstract

**Simple Summary:**

Bumble bees have annual colonies that produce new queens at the end of summer. They mate and accumulate substantial fat reserves before overwintering underground prior to emerging in the spring. Because body size influences how much fat a queen can accumulate in the fall and how quickly she burns through those store in the winter, we asked if queen body condition influences the timing of spring emergence. We measured mass and body condition (indicative of remaining fat reserves) of bumble bee queens in the spring. Large queens and those in better body condition emerged throughout the spring, whereas lighter queens and those with poorer body condition emerged earlier, suggesting that depleted fat reserves could trigger early emergence of bumble bee queens.

**Abstract:**

For insects, the timing of many life history events (phenology) depends on temperature cues. Body size is a critical mediator of insect responses to temperature, so may also influence phenology. The determinants of spring emergence of bumble bee queens are not well understood, but body size is likely important for several reasons. In fall, queens accumulate energy stores to fuel overwinter survival. Accumulation of fat stores prior to and depletion of fat stores during overwintering are likely size-dependent: larger queens can accumulate more lipids and have lower mass-specific metabolic rates. Therefore, larger queens and queens in relatively better condition may have delayed depletion of energy stores, allowing for later spring emergence. To test whether timing of spring emergence is associated with body size and condition, we captured 295 *Bombus huntii* queens in Laramie, WY, during the 2020 and 2021 growing seasons, weighed them, and measured intertegular width (a size metric unaffected by variation in feeding and hydration state). Early emerging queens were smaller than later emerging queens across years. Mass relative to intertegular width increased as the season progressed suggesting, as predicted, that body condition influences the timing of spring emergence for these crucial pollinators.

## 1. Introduction

Temperature is a major factor influencing the timing of critical life history events for diverse organisms (“phenology”; [1]). Spring emergence, peak abundance, and fall immergence (e.g., entry into diapause, overwintering quiescence, or hibernation) have all been linked to temperature cues [2,3,4], and phenological shifts are well-documented responses to ongoing climate change. Yet, phenological responses of diverse taxa to changing climates often vary considerably in magnitude and direction [5,6]. At broad spatial scales, drivers of phenological shifts appear to vary with latitude; temperature is a strong driver at higher latitudes but becomes increasingly less important at tropical latitudes where phenological shifts are tied more closely to precipitation [7]. On local scales, some organisms emerge later when snowmelt is delayed [8], whereas others emerge earlier in the spring despite increased snowpack [9]. Such variation in phenological responses to changing climate may reflect differences between the macroclimates used to assess responses and the microclimates where organisms live [10,11,12]. Further, differential physiological responses to temperature and other cues may underly variability in phenological responses [13].

Some work suggests that microclimatic variation in temperature can strongly alter phenology over small spatial scales. Flowers on south-facing slopes bloom as much as eleven days earlier than those of the same species found only 50 m away on north-facing slopes [14]; and in insects, ash borer beetles (*Agrilus planipennis*) 30 mm under the south-facing bark of urban ash trees (*Fraxinus*) are predicted to develop up to 30 days faster, thereby advancing emergence relative to those exposed to cooler and more variable air temperatures [15]. Even within the same site or microclimate, variation in phenology has been linked to physiological characteristics, such as sex and reproductive strategy (i.e., time of season when mating occurs; [16,17]). Phenological shifts can also depend on age and gravidity. In little brown bats (*Myotis lucifugus*), mature females emerge from hibernation earlier than both younger females and males [17]. In Richardson’s ground squirrels (*Spermophilus richardsonii*), entry into hibernation for females varies with rate of weight gain which can be influenced by recovery from birthing and rearing of offspring [18], providing evidence that the physiological state of an individual can alter the timing of critical life events even within the same life stage.

Body size is a fundamental trait affecting physiology and ecology that may also influence timing of spring emergence. Across taxa, overwintering survival is often positively correlated with body size due in large part to links between size and both total lipid stores and rate of lipid depletion [19,20,21]. Body size mediates ectotherm responses to temperature, and thus likely also mediates phenological responses [22,23]. Across taxa, phenological responses vary with body size, with some organisms emerging earlier as size increases [16,21,24] while others emerge later; for some, phenology does not vary with body size. For example, in midges (*Chironomidae*) and damselflies (*Zygoptera*), body size (wing length) decreased with emergence date [25,26]. In dragonflies (*Anisoptera*), body size was not linked to timing of emergence [26]. Body mass and condition can also be important determinants of emergence timing [27]. In mason bees (*Osmia cornuta*), body mass was not linked to the timing of emergence [28], yet often if body mass does not influence emergence timing, body condition does [27], and vice versa [16,17]. Individuals that emerge early risk exposure to fatally cold temperatures while those that emerge late may face increased competition for resources, but research on the influence of body size and more so condition on timing of emergence remains surprisingly sparse.

Queen bumble bees eclose in late summer and fall, acquire substantial lipids stored in their fat body, mate, and then overwinter underground before emerging the following spring [29,30] while the remaining maternal colony dies. In spring, newly emerged queens feed on nectar and pollen as their ovaries develop [31] and eventually find suitable sites (usually underground) to start new colonies. The timing of spring emergence of bumble bee queens is clearly, in part, related to temperature [32], but can vary strikingly even for the same species at the same site [33,34,35] with important implications: those that emerge earlier may risk exposure to spring cold snaps [13] with increased likelihood of mortality [33,36] while those that emerge later may face increased competition for suitable nest sites and floral resources [37,38].

We aimed to address one factor that may lead to pronounced variation in the timing of spring emergence of bumble bee queens. Given that they are responding to temperature cues, differences in overwintering microclimates may, in part, lead to different emergence times [16,39,40,41]. Body size may also play an important role given potential effects of size on accumulation of lipid stores (larger queens can store more lipids; [20]), and on mass-specific metabolism [42,43,44] which should mean that larger queens deplete lipid stores more slowly. If queens must emerge before fully depleting stored lipids, we would predict that smaller queens would emerge earlier because, given the same overwintering temperatures, they would deplete lipid reserves more quickly (both due to smaller initial stores and higher overwintering metabolic rates). Similarly, regardless of size, we would predict that queens with poorer body condition (i.e., smaller lipid reserves) would emerge earlier. 

To test these predictions, we measured mass and intertegular width (ITW, a measurement of exoskeletal size fixed at eclosion; [45,46]) of *Bombus huntii* queens throughout the spring emergence period for two years in Laramie, WY. Queens that weighed less emerged earlier as predicted. We used the comparison between mass and ITW to infer body condition (BeeMI) and found a striking pattern: queens that were relatively light for their size (in poor condition) emerged earlier, dominating the early spring emergence peak, whereas those in good condition (relatively heavy for their size) dominated the late spring emergence peak. 

## 2. Materials and Methods

### 2.1. Animal Collection

Throughout the 2020 and 2021 growing seasons, we captured *B. huntii* queens by net once a week in Laramie, WY (2188 m; 41.316, −105.586 +/− 0.5 mi), starting a few days after the first bumble bee was seen (30 April 2020 and 2021) and continuing until queens were no longer captured during a collection event (Figure 1). To minimize possible effects of weather conditions on sampling, we selected days and sampling periods with the best conditions for bumble bee activity (warm, sunny). Collections were standardized for 3 person-hours, and terminated early only when 50 total bees (queens or workers) were collected. Once captured, each queen bee was kept in a ventilated vial on ice for transport to the lab. 

### 2.2. Body Size and Condition

Immediately following each survey, bees were weighed to the nearest mg (Acculab ALC-210.4; Sartorius Group, Göttingen, Germany). In 2021, bees were then photographed from the dorsal view next to an object of known size and released. The width between the tegulae (intertegular width, ITW, mm, also termed intertegular span, ITS; [45]) was measured from photographs by first setting the scale based on the object of known size and then measuring the length of a straight line drawn between the outside edges of the tegula using ImageJ [47,48]. ITW is not affected by variation in feeding or hydration state, so is a reliable estimate of body size of bumble bees [45,46,49]. We estimated body condition of queens as the bee mass to ITW ratio (BeeMI). Higher BeeMI indicates better body condition under the assumption that variation in mass of queens with the same exoskeletal dimensions is due primarily to the mass of lipid stores (for bumble bee queens prior to nest initiation, growth and depletion of the fat body is the dominant determinant of variation in mass; [50,51,52]).

### 2.3. Statistical Analyses

After visual inspection for normality, we compared ordinary least squares (OLS) and standard major axis (SMA) regressions of mass on ITW using package lmodel2 [53] in R ([54]; see Appendix A, Figure A1 and Figure A2). Based on visual inspection of plots and regression outputs, SMA was used to predict mass from ITW (R^2^ = 0.249; *n* = 165). Predicted values were extracted from the fitted lines, with those above and below the predicted lines heavier and lighter than expected, respectively, based on exoskeletal size (ITW) fixed at eclosion (Figure 2). 

We confirmed that mass, ITW, and body condition (BeeMI) did not deviate strongly from normality by visual inspection of residuals, Q-Q plots, and histograms. To account for nonlinearity, we modeled the dependence of emergence time (day of year) on mass, ITW, and body condition using generalized additive models (GAMs) using package mgcv() [55] in R [54]. Overfitting of unconstrained models limited biologically relevant inferences. Therefore, to limit overfitting, we used a cubic regression smoother and, given 11 survey days in each year, we limited smoothers to 3 knots to constrain the variance of the data and better explain the biological relevance. Models with knots constrained to 3 were included in Table 1 (indicated by k = 3), which is reflected in the effective degrees of freedom (edf) for each model; an edf of 1 indicates a linear relationship while an edf greater than 1 indicates a strongly nonlinear relationship that increases with the edf value [56]. We compared linear models and GAMs with and without year as a covariate using Akaike information criterion values (AIC), and compared AIC values of models only where k = 3 was indicated in final model selection.

To further assess the relationship between body condition and emergence timing, we used a chi-square contingency table to analyze the association between body condition categories (heavier or lighter than expected) and the two clear emergence peaks.

## 3. Results

In total, we captured 132 queens between April and July of 2020 and 165 queens between April and July of 2021. Although queens were captured each week throughout the span of the 2+ month spring emergence period, they emerged in two distinct waves in both years (Figure 1). The bimodal distribution of spring emergence occurred during the same weeks each year despite large variation in weather conditions; a cold snap brought several inches of snow on 8 June 2020, whereas no late-spring cold snaps hit Laramie in 2021. Peak queen abundance was on 19 and 18 May (days 140 and 138) in 2020 and 2021, respectively, with the second peak occurring on 18 and 16 June (days 170 and 167) in 2020 and 2021.

Masses from 235 individuals total were collected across both years (70 queens in 2020 and 165 queens in 2021). The GAM including year as a covariate and the interaction between year and day of year had the lowest AIC value and was selected as the best fitting model (Table 1). Mass of bumble bee queens increased significantly with day of emergence for both years, accounting for over 39% of the variation; mass increased rapidly at the beginning of the emergence period, then tapered off (GAM, R^2^ = 0.397, df = 6.770; *n* = 295; Figure 3A). When modeling the relationship between ITW and emergence timing, the GAM without knot limitations had the lowest AIC score, but the GAM with knots limited to three was selected as the best fitting model given the small sample size (Table 1). ITW varied nonlinearly, increasing with day of emergence though this trend was statistically insignificant (GAM, R^2^ = 0.0794, df = 3.448, *n* = 165; Figure 3B). Much like mass, body condition (mass/ITW) increased nonlinearly as spring emergence progressed. We compared GAMs describing the relationship between body condition and emergence timing and selected the model with knots limited to 3 given the small sample size (Table 1). Early emerging queens had the lowest body condition (low BeeMI) with later emerging queens having increasingly better body condition until early June, at which point body condition plateaued (GAM, R^2^ = 0.456, df = 3.958; *n* = 165). While mass was positively correlated to emergence timing, body condition better accounted for the timing of spring emergence and explained over 45% of the variation of queen emergence timing (Figure 3C).

After categorizing queen body condition as heavier or lighter than expected based on the regression estimate (Figure 2), emergence timing was strongly associated with relative body condition for queen bees. The first emergence peak was dominated by queens that were lighter than expected, while heavier than expected queens made up a larger fraction of the second emergence peak (χ^21^ = 17.87, *p* < 0.0001, *n* = 165; Figure 4A). Queens of lower body condition emerged rapidly early on and then tapered off, whereas queens with better body condition emerged overall later with rate of emergence increasing later in the season (Figure 4B).

## 4. Discussion

Mass alone explained 40% of the variation in timing of queen emergence (Figure 3A; Table 1). While body mass influences overwintering survival in other bee species (*Osmia*), it is sex-specific—larger male mason bees had higher rates of survival during overwintering than smaller males, but this pattern did not hold true for females nor did it influence timing of emergence [28]. For commercially raised bumble bee queens (*B. terrestris*) held in cold storage, those less than 0.6 g had high mortality regardless of temperature or duration of storage. Queens larger than 0.6 g had higher mortality when held for longer durations, regardless of the cold storage temperature [57]. Therefore, it may be that the smallest *B. huntii* queens died; for those that emerged, it is clear that timing of emergence is tied to mass. 

Mass is often used as a proxy for body condition. Our simple metric of body condition (BeeMI) explained even more (over 45%) of the variation in phenology of queen bumble bees. Relatively lighter queens emerged earlier (Figure 4), providing evidence that timing of bumble bee queen emergence is coupled to body condition. While previous studies have recorded increased survival of queen bumble bees that emerged later in the season, body condition was not assessed [36]. We are unaware of work assessing patterns of this sort in other insects, but other animals have shown the converse pattern: for both rattlesnakes (*Crotalus viridis viridis*) and arctic ground squirrels (*Spermophilus parryii kennicottii*), relatively heavier individuals emerged earlier than lighter ones [16,27]. This contrasting pattern may be due to other reproductive pressures; for rattlesnakes, early emergence increases the time available for mating [16] and, for squirrels, early emergence increases offspring growth and survival [27]. Bumble bees mate in the fall so mating probability is decoupled from spring emergence. Only the first brood of offspring (which eclose in roughly three weeks; [58]) is dependent on the queen for resources as subsequent broods are provisioned by their sisters. As such, colony growth may depend less on timing of spring emergence and more on the timing of local resource availability [8]. 

Queens of similar body condition still varied in timing of spring emergence (Figure 3C). This variability may be linked to the timing of when they actually emerged. For example, a captured queen may have emerged that morning or a week or two prior such that they may differ in ovary development or body condition. Additionally, variability may be linked to selection of overwintering sites which, due to differences in microclimatic conditions, may influence spring emergence and survival [14,59]. Bumble bees overwinter underground [60] where, dependent on depth and soil characteristics, they may encounter strikingly different temperatures [61]. Therefore, where queens overwinter could alter when they experience temperatures that trigger spring emergence. Characterizing overwintering sites in the field and measuring how body size influences accumulation and depletion of energy stores in the fall and winter, respectively, could reveal key drivers of phenology of queen bumble bees and facilitate predictions of climate change impacts on these key pollinators. 

Timing of emergence was not significantly linked to ITW (Figure 3B), suggesting that condition, not fixed size (exoskeletal size at eclosion) per se, is a key determinant of timing of spring emergence. Nonetheless, measurements of ITW and mass for the same individual provide a straightforward approach to estimate body condition for bumble bees (Figure 2) and possibly other insects as well. Such estimates of body condition may prove useful not only in phenology studies but in studies on land use effects [62], habitat changes [63], agriculture [64], and conservation [65,66] on health of insect populations. 

A bimodal distribution of emergence is apparent across both years (Figure 1). Queen *B. huntii* emerged over two months, a range witnessed in previous studies. Emergence timing varies with species: *B. terrestris* emerge before *B. lapidarius* lasting for two to three months, though duration and timing varies across years [33]. During our surveys, however, the timing of *B. huntii* emergence from start to end occurred in two distinct waves that overlapped surprisingly closely with day of year across both years. These abundance peaks occurred within three days of each other across two years despite differences in weather conditions, indicating that emergence cues that overwintering queens are responding to may not be as strongly coupled to acute variation in air temperatures or weather than other factors. Photoperiod can influence phenology of some insects (reviewed in [4]), but this is likely not the case for bumble bees as they overwinter underground, sheltered from sunlight cues. Aside from external factors, internal factors (e.g., gene expression) may influence timing of spring emergence resulting in the bimodal peaks of emerging queens [67]. 

In addition to temperature being a major driver of phenology in bumble bees [8,32,34], our findings highlight the importance of body condition in bumble bee phenology, with queen physique tightly linked to timing of emergence in the spring. With over half of their lives spent overwintering underground and emergence spanning over two months (Figure 1), the physical condition of queens directly affects whether they emerge earlier or later in the season, potentially altering their already short active season by weeks. Variation in lipid accumulation during a small window in the fall [20,68] and in lipid depletion due to overwintering site selection likely contributes substantially to the timing of queen emergence in the spring. As bumble bee phenology shifts in response to changing climates with the most pronounced responses occurring in the last 40 years [32], further research characterizing the impact of emergence timing on colony success may uncover cascading effects of this critical life history stage on bumble bee populations. 

## Figures and Tables

**Figure 1 insects-13-00870-f001:**
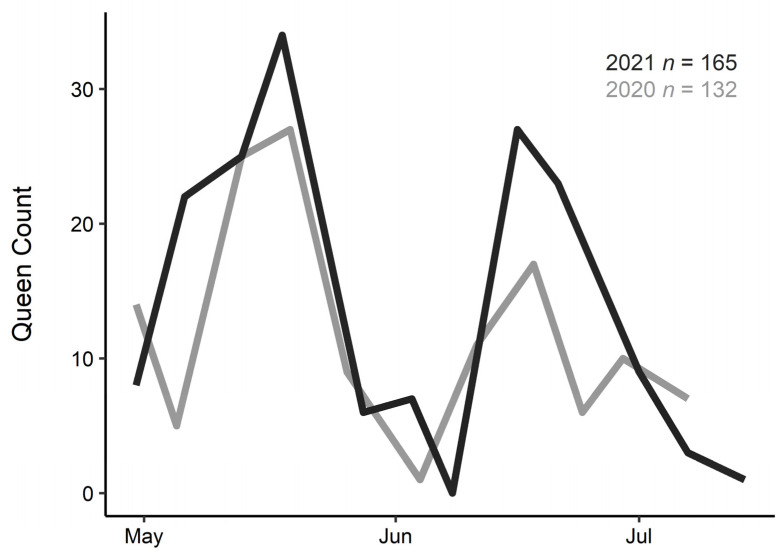
Bumble bee queens (*B. huntii*) emerged across two and a half months during the spring with a marked bimodal distribution across both 2020 (grey; *n* = 132) and 2021 (black; *n* = 165).

**Figure 2 insects-13-00870-f002:**
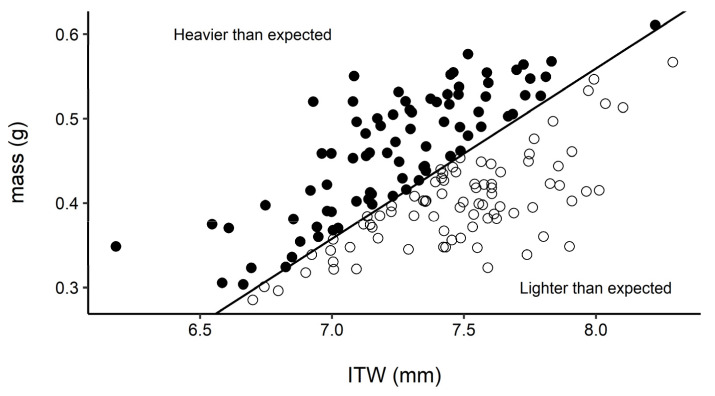
Body mass generally tracks intertegular width for bumble bee queens, with variability in mass at a given ITW indicative of body condition. Queens above the SMA regression line (R^2^ = 0.249) were heavier than expected (solid circles) and those below were lighter than expected (empty circles) given exoskeletal size fixed at eclosion. ITW measurements were only available for 2021.

**Figure 3 insects-13-00870-f003:**
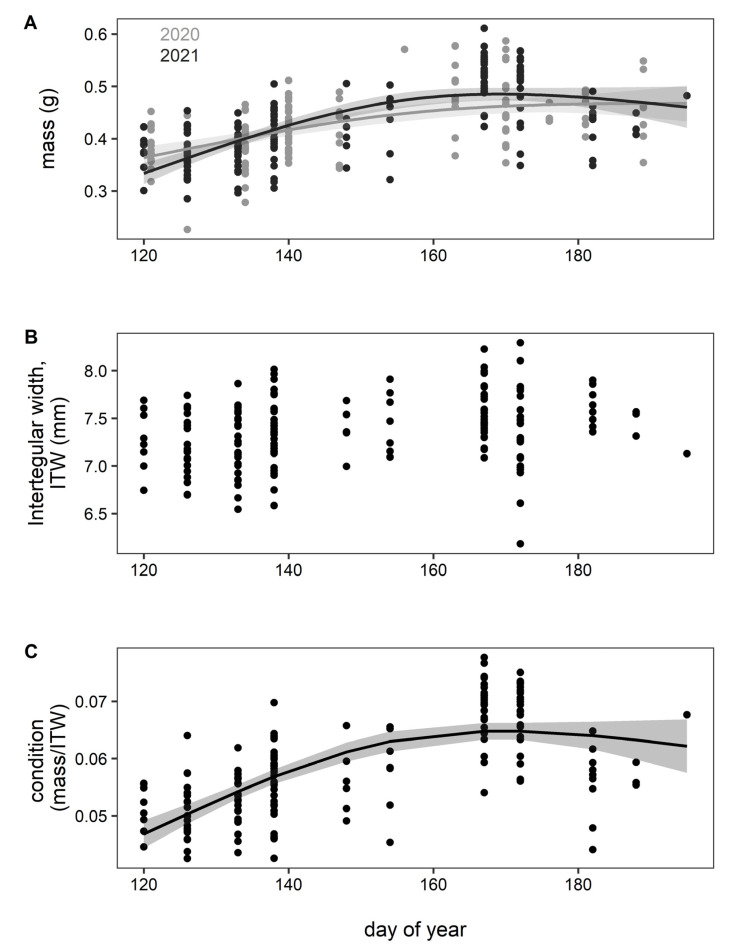
Mass and body condition varied with timing of queen emergence, while ITW did not. (**A**) Queen mass increased significantly with day of emergence for both 2020 (gray points and lines; *n* = 130) and 2021 (black points and lines; *n* = 165) (GAM, R^2^ = 0.397, df = 6.770; *n* = 295; Table 1); (**B**) ITW did not change significantly with day of year (GAM, R^2^ = 0.0794, df = 3.448, *n* = 165; Table 1); (**C**) Body condition (as estimated by the ratio of mass to ITW) increased strikingly with day of queen emergence early in the season, with day of emergence explaining over 45% of the variation in body condition (GAM, R^2^ = 0.456, df = 3.958, *n* = 165; Table 1).

**Figure 4 insects-13-00870-f004:**
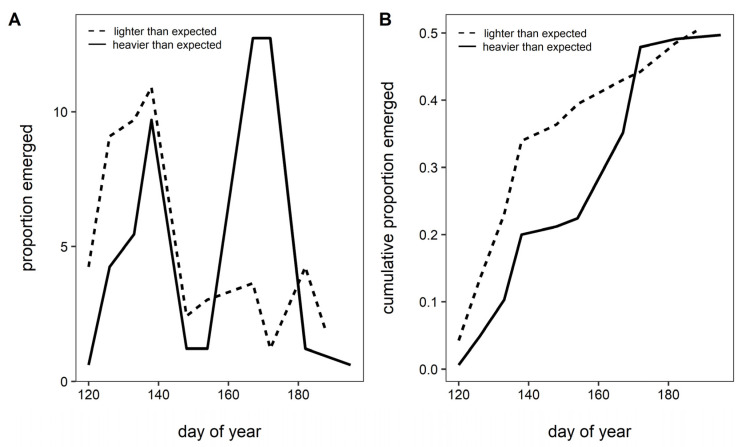
Queen body condition was strongly associated with the two queen emergence peaks. (**A**) Relatively light queens (dashed lines) dominated the first emergence peak, and (**B**) tended to emerge rapidly early in the season. Conversely, relatively heavy queens (solid line) emerged during both peaks, but dominated the second emergence peak, when lighter than expected queens were scarce (A; χ^21^ = 17.87, *p* < 0.0001, *n* = 165).

**Table 1 insects-13-00870-t001:** GAM outputs and AIC values from analyzing mass, ITW, and body condition with emergence timing. Aside from the initial model for each response variable, we limited smoothers to 3 knots to better explain the biological relevance and fit smoothers with a cubic regression to prevent overfitting. Best fit models are in bold.

Response Variable	Model	AIC	ΔAIC	R^2^	Deviance Explained	edf	df	*n*
**mass**	**~s (day of year, k = 3) + year + s (day of year) × year**	**−889.406**	**0**	**0.397**	**40.70%**	**2020:1.821 2021:1.950**	**6.770464**	**295**
**mass**	~s (day of year, k = 3)	−888.142	1.264	0.389	39.30%	1.958	3.958283	295
**mass**	~s (dayofyear, k = 3) + year	−887.086	2.32	0.389	39.50%	1.958	4.958068	295
**ITW**	**~s (day of year, k = 3)**	**116.9413**	**0**	**0.0794**	**8.76%**	**1.448**	**3.447957**	**165**
**condition**	**~s (day of year, k = 3)**	**−1195.27**	**0**	**0.456**	**46.30%**	**1.958**	**3.957785**	**165**

## Data Availability

The data used in this study are available by email request to the corresponding author.

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
