# Peer review of "Phat Queens Emerge Fashionably Late: Body Size and Condition Predict Timing of Spring Emergence for Queen Bumble Bees"

_insects, 2022, doi:10.3390/insects13100870_

Round 1

Reviewer 1 Report

The manuscript is well written and reports on interesting findings that will appeal to the broad readership of the journal Insects. The findings and discussion is useful for anyone working on seasonal emergence of Hymenoptera with respect to body size and condition (for example emergence of Vespula spp. queens).

I commend the authors for being able to record these data without destructively sampling the study species! This means this work could also be done for species that are more conservation sensitive.

My only criticism has to do with the description of the General Additive Modelling, and how the best model was selected that was done for bumble bee mass. It is not clear if you also used R for this part of the study and how regression smoothing was controlled. With the results in Table 1, your best fitting model did not have the lowest AIC, although this is what is stated in the Methods. Typically, change in ∆AIC is reported to help prove selection of on model over another. The non-interaction model was very close to your much more complex model that included interactions. In Table 1 what does ‘ edf ’ stand for? Should d.f. not be non-decimal?

Minor comments:

Consider shortening the third paragraph of the Introduction. With the numerous examples given it slightly breaks the flow.

Use superscript where appropriate, e.g., R2 or χ21

Figure A1 – what does the red and two grey lines represent?

Consider adding a photo or two of the study species and how individual queens were measured/weighed. I looked up the species on Google and saw that it had an interesting colour pattern that was not mostly black (not being familiar with this particular species). Presumably this would also affect its thermal properties when foraging.

Author Response

(Reviewer comments in italics, responses in bold)

Comments and Suggestions for Authors

The manuscript is well written and reports on interesting findings that will appeal to the broad readership of the journal Insects. The findings and discussion is useful for anyone working on seasonal emergence of Hymenoptera with respect to body size and condition (for example emergence of Vespula spp. queens).

I commend the authors for being able to record these data without destructively sampling the study species! This means this work could also be done for species that are more conservation sensitive.

We appreciate the comments and agree! We hope this nondestructive approach could be useful for many non-Hymenopteran insects as well.

My only criticism has to do with the description of the General Additive Modelling, and how the best model was selected that was done for bumble bee mass. It is not clear if you also used R for this part of the study and how regression smoothing was controlled. With the results in Table 1, your best fitting model did not have the lowest AIC, although this is what is stated in the Methods. Typically, change in ∆AIC is reported to help prove selection of on model over another. The non-interaction model was very close to your much more complex model that included interactions. In Table 1 what does ‘ edf ’ stand for? Should d.f. not be non-decimal?

We greatly appreciate your detailed feedback! Decimals were removed from df. To describe how we modeled emergence time on mass, ITW, and body condition, we added that we used “…package mgcv() [55] in R [54].” on line 173. To clarify how we controlled for smoothing as well as defining what edf stands for, we added the following to line 175:

“Overfitting of unconstrained models limited biologically relevant inferences. Therefore, to limit overfitting, we used a cubic regression smoother to prevent overfitting and, given 11 survey days in each year, we limited smoothers to 3 knots to constrain the variance of the data and better explain the biological relevance.  even though more knots increased the model fit. Models with knots constrained to 3 were included in Table 1 (indicated by k=3), which is reflected in the effective degrees of freedom (edf) for each model; an edf of 1 indicates a linear relationship while an edf greater than 1 indicates a strongly nonlinear relationship that increases with the edf value [56]. We compared linear models and GAMs with and without year as a covariate using Akaike information criterion values (AIC), and compared AIC values of models only where k=3 was indicated in final model selection.”

We removed the unconstrained models of mass, ITW, and condition from Table 1, including only those where knots were limited to 3, placing the best fitting model at the top. The ∆AIC column was also added as suggested.

Minor comments:

Consider shortening the third paragraph of the Introduction. With the numerous examples given it slightly breaks the flow.

We shortened the third paragraph by removing several examples of taxa beyond insects.

Use superscript where appropriate, e.g., R2 or χ21

We applied superscript changes to lines 167, 229, 230, 233, 238, and 247.

Figure A1 – what does the red and two grey lines represent?

To describe the red and grey lines, we added the following information to Figure A2 (line 350):  “with red lines indicating the OLS (left) and SMA (right) regression lines and grey lines indicating the 95% confidence intervals of each slope.”

Consider adding a photo or two of the study species and how individual queens were measured/weighed. I looked up the species on Google and saw that it had an interesting colour pattern that was not mostly black (not being familiar with this particular species). Presumably this would also affect its thermal properties when foraging.

We added in a figure in the appendix (Figure A1) to show how a B. huntii queen in the field, and how we weighed and photographed individuals to determine body condition. This figure can be placed as a supplemental figure or a figure in the main document if you and the other reviewers would prefer.

Reviewer 2 Report

Manuscript:  insects-1915166-peer-review-v1

Phat queens emerge fashionably late: body size and condition 2 predict timing of spring emergence for queen bumble.  bees  Ellen C. Keaveny  and Michael E. Dillon

The authors wanted to address the timing of springtime emergence of bumblebee queens (Bombus huntii), to body condition of the queens. The authors have collected in two years an impressive number of queens. They concluded that the queens emerged in two peaks in spring and that mass and body condition is related to the emerging date, as queens with a lower mass and body condition emerge meanly early in spring and the heavier queens later in spring, which are very interesting results. Also is concluded that body size is not related to emerging time.

The authors have interesting results and is worth to publish, if more information is given about the temperatures during the spring period, flowering and  more in depth discussion related to the biology of the queens in relation to the phenology.

Introduction

-As the authors did research on bumblebees, an insect, examples of non-insects (the bats, snakes and ground squirrels) are less relevant as they have a different physiology. The authors have given enough information about other insects.

-In r 90 the authors start with bumble bee without a short explanation of bumblebees. For instance the authors can start with: “Bumblebees live in annual colonies. Young bumblebee queens feed in the colony on pollen and develop a fat body with lipid reserve to survive, alone, a long period of diapause”.

-In r 94 the authors indicate that the emergence of queens in spring is related to temperature. It is important that the authors give an explanation (on basis of literature) why it would be relate to temperature.

-In r109 the authors give their definition: body condition=lipid reserves relative to their fixed body size. In the Methods r 141-144 it is explained better with references. It is needed to replace this to the Introduction as it is a very important assumption, as body condition is used 36 times.

Methods

It is important to show also the weather conditions (temperatures) during the catching period, and also before the first queens emerges.

I expect that the authors have data about the caught queens if they have filled pollen baskets and on which flowers they are caught, or which plant are flowering that are of interest of bumble bee queens.

Results

Figure 3 show interesting results. As indicated by the authors the queens that are caught in early spring have a lower mass/body size, and the large group that emerge in the late spring have a significant higher mass/body size. However, it can also be see in the figure that the queens that are caught later in early spring and later in late spring have the same mass/body size. The question is  what kind of queens are they. These queens could have fly for a longer time, searching for a nest and  losing their lipid reserve and develop their ovaries. This must be discussed.

Discussion

-r229 Why no literature about bumblebee?  The authors could refer to the paper of Beekman et al, 1998.

-r280 The sentence start with ‘In addition to temperature…..’. In the rules 271-275 is however discussed that temperature is less important. That seems contra dictionary, ánd no data are given about temperature. It must be not difficult to get these data.

-the authors refer several times to climate change and phenology, but not discussed it the relation with flowers. For queens it is useless to emerge if there are no flowers, and flowering of spring flowers is related to temperature. It would be interesting if the authors show data and discuss which plants are flowering during the two peaks.

Author Response

(Reviewer comments in italics, responses in bold)

Response to Reviewer #2

Comments and Suggestions for Authors

Manuscript:  insects-1915166-peer-review-v1

The authors wanted to address the timing of springtime emergence of bumblebee queens (Bombus huntii), to body condition of the queens. The authors have collected in two years an impressive number of queens. They concluded that the queens emerged in two peaks in spring and that mass and body condition is related to the emerging date, as queens with a lower mass and body condition emerge meanly early in spring and the heavier queens later in spring, which are very interesting results. Also is concluded that body size is not related to emerging time.

The authors have interesting results and is worth to publish, if more information is given about the temperatures during the spring period, flowering and  more in depth discussion related to the biology of the queens in relation to the phenology.

Introduction

-As the authors did research on bumblebees, an insect, examples of non-insects (the bats, snakes and ground squirrels) are less relevant as they have a different physiology. The authors have given enough information about other insects.

We removed several examples referencing other taxa in the third paragraph of the introduction but left them as citations to leave as examples in how our research question and approach are widely applicable to organisms beyond bumble bees and insects.

-In r 90 the authors start with bumble bee without a short explanation of bumblebees. For instance the authors can start with: “Bumblebees live in annual colonies. Young bumblebee queens feed in the colony on pollen and develop a fat body with lipid reserve to survive, alone, a long period of diapause”.

To better introduce bumble bee life history, we added “acquire substantial lipids stored in their fat body,” to line 99 and “while the remaining maternal colony dies” to line 101.

We also altered the Simple Summary to provide background information on queen bumble bee life history.

-In r 94 the authors indicate that the emergence of queens in spring is related to temperature. It is important that the authors give an explanation (on basis of literature) why it would be relate to temperature.

The correlation between spring temperatures and emergence of bumble bees is well established and we cite several of the most relevant papers, including references 8, 32, and 34. The mechanisms by which spring temperatures trigger queen emergence have not been extensively studied to our knowledge. We are also very interested in those mechanisms, but they are outside the scope of this work which aims to test whether other factors could account for variability in spring emergence at the same site where local temperatures are likely fairly consistent.

-In r109 the authors give their definition: body condition=lipid reserves relative to their fixed body size. In the Methods r 141-144 it is explained better with references. It is needed to replace this to the Introduction as it is a very important assumption, as body condition is used 36 times.

We would argue that body condition is a general concept, understandable without a complete description of the approach used to measure it; the details of the approach are appropriately desribed in the Methods of the manuscript. That point aside, the text in the last paragraph of the introduction provides our approach to estimating body condition so that the reader can, at that point, already understand the result:

            “To test these predictions, we measured mass and intertegular width (ITW, a measurement of exoskeletal size fixed at eclosion; [45,46]) of Bombus huntii queens throughout the spring emergence period for two years in Laramie, WY. Queens that weighed less emerged earlier as predicted. We used the comparison between mass and ITW to infer body condition (BeeMI) and found a striking pattern: queens that were relatively light for their size (in poor condition) emerged earlier, dominating the early spring emergence peak whereas those in good condition (relatively heavy for their size) dominated the late spring emergence peak.

Methods

It is important to show also the weather conditions (temperatures) during the catching period, and also before the first queens emerges.

This is a great point that we didn’t clearly explain. We’ve added the following:

“To minimize possible effects of weather conditions on sampling, we selected days and sampling periods each that had the best conditions for bumble bee activity (warm, sunny).”

I expect that the authors have data about the caught queens if they have filled pollen baskets and on which flowers they are caught, or which plant are flowering that are of interest of bumble bee queens.

While we have some data describing the flower type that the queens were foraging on and presence or absence of pollen baskets, however, these data were not consistently recorded during every survey preventing use to including it in our analyses. Although it would be a fun research avenue to travel down if we had that data on hand, we argue that describing flower type throughout the growing season is not relevant to our main research question and would not alter our main findings.

Results

Figure 3 show interesting results. As indicated by the authors the queens that are caught in early spring have a lower mass/body size, and the large group that emerge in the late spring have a significant higher mass/body size. However, it can also be see in the figure that the queens that are caught later in early spring and later in late spring have the same mass/body size. The question is  what kind of queens are they. These queens could have fly for a longer time, searching for a nest and  losing their lipid reserve and develop their ovaries. This must be discussed.

We agree that this is an interesting pattern. On lines 275, we comment on this, describing the observation a bit differently: “Queens of similar body condition still varied in timing of spring emergence (Figure 3C).”  Beyond the discussion of possible microclimate effects, we added the following:

“This variability may be linked to the timing of when they actually emerged. For example, a captured queen may have emerged that morning or a week or two prior such that they may differ in ovary development or body condition.”

This would be a neat target for future studies, but it is unclear to us how to infer how long it had been since field-collected queens have emerged.

Discussion

-r229 Why no literature about bumblebee?  The authors could refer to the paper of Beekman et al, 1998.

Great point—while we wanted to keep the discussion broad and applicable across taxa, it is important to better tie our discussion back to the focal organism. Beekman et al. (1998) kept commercially-reared queens in cold storage at factorial combinations of temperature and duration and measured survival and post-cold storage egg-laying. However, their finding that smaller queens tend to have high mortality provides further support for our hypothesis that successful overwintering and timing of spring emergence are contingent on fat stores. We added the following statement citing Beekman to line 253:

“For commercially-raised bumble bee queens (B. terrestris) held in cold storage, those less than 0.6 g had high mortality regardless of temperature or duration of storage. Queens larger than 0.6 g had higher mortality when held for longer durations, regardless of the cold storage temperature [58]. Therefore, it may be that the smallest B. huntii queens died; for those that emerged, it is clear that timing of emergence is tied to mass.”

-r280 The sentence start with ‘In addition to temperature…..’. In the rules 271-275 is however discussed that temperature is less important. That seems contra dictionary, ánd no data are given about temperature. It must be not difficult to get these data.

The correlations between spring temperatures and emergence of bumble bees is well established and we cite several of the most relevant papers, including references 8, 32, and 34.  On line 303, we argue that emergence cues may not be linked to “acute variation in air temperatures or weather” (i.e., short term shifts in air temperatures). These are distinct from temperatures in the ground, where bumble bees are overwintering; ground temperatures are likely largely buffered from shifts in air temperatures associated with, for example, weather fronts. Nevertheless, ground temperatures do change seasonally and thus likely, in part, cue bumble bee emergence.

-the authors refer several times to climate change and phenology, but not discussed it the relation with flowers. For queens it is useless to emerge if there are no flowers, and flowering of spring flowers is related to temperature. It would be interesting if the authors show data and discuss which plants are flowering during the two peaks.

Overlapping phenology of flowers and bumble bees is critical, and the possibility of phenological mismatches is an active and exciting area of research! However, the focus of this work is on whether intrinsic cues may be triggering spring emergence of queen bumble bees (which most likely occur in the absence of any floral cues, given that they are underground prior to emergence). Because floral relationships were not the focus of this work, we don’t have data on flowers, but we agree it would be an interesting avenue for future work.

Reviewer 3 Report

Mild changes in methodology description are needed. (For example the average time of the day when the catching took place.)

It’s not clear to me how authors know that the queen is nearly emerged. I assume Bumble bee queen fly, till the first generation of workers is born. Therefore it could be possible to catch a queen that emerged 14 days back.

How was estimated the amount of nectar inside intestinal tract of queens? (Honeybee worker can carry 60 % of body weight in their honey stomach.) It seem to me that this variable could influence  your results. 

Author Response

(Reviewer comments in italics, responses in bold)

Response to Reviewer #3

Comments and Suggestions for Authors

Mild changes in methodology description are needed. (For example the average time of the day when the catching took place.)

To describe the conditions while sampling, we added the following to line 134:

“To minimize possible effects of weather conditions on sampling, we selected days and sampling periods each that had the best conditions for bumble bee activity (warm, sunny).”

It’s not clear to me how authors know that the queen is nearly emerged. I assume Bumble bee queen fly, till the first generation of workers is born. Therefore it could be possible to catch a queen that emerged 14 days back.

We do not know if individual queens were newly emerged or not and avoided making predictions. It is possible that a flying queen we captured had emerged 14 days prior. To account to the variation in emergence across captures we added the following to line 276:

“This variability may be linked to the timing of when they actually emerged. For example, a captured queen may have emerged that morning or a week or two prior such that they may differ in ovary development or body condition.”

How was estimated the amount of nectar inside intestinal tract of queens? (Honeybee worker can carry 60 % of body weight in their honey stomach.) It seem to me that this variable could influence  your results.

We addressed this briefly in the methods but agree the manuscript would be strengthened by following up in the Discussion. We did not measure crop contents. However, as we discuss in the methods, “for queen bumble bees prior to nest initiation, growth and depletion of the fat body is the dominant determinant of variation in mass.” Variation in crop contents may contribute some to the variation in mass seen in Figure 2, but the magnitude is likely quite small relative to changes due to depletion of fat stores. Worker bumble bees can similarly have large increases in body mass associated with nectar loads but queens largely feed themselves with a little extra for moistening pollen balls on which larvae feed.

Round 2

Reviewer 2 Report

There are no new comments on the revised manuscript.